# Goal-constrained planning domain model formal verification of safety properties*

***
***
***

## Abstract

The verification of planning domain models is crucial to ensure the safety, integrity and correctness of planning-based automated systems. This task is usually performed using model checking techniques. However, directly applying model checkers to verify planning domain models can result in false positives, i.e. counterexamples that are unreachable by a sound planner when using the domain under verification during a planning task. In this paper, we discuss the downside of unconstrained planning domain model verification. We then propose a fail-safe practice for designing planning domain models that can inherently guarantee the safety of the produced plans in case of undetected errors in domain models. In addition, we demonstrate how model checkers, as well as state trajectory constraints planning techniques, should be used to verify planning domain models so that unreachable counterexamples are not returned.

## 1 Introduction

Planning and task scheduling techniques are increasingly applied to real-world problems such as activity sequencing, constraint solving and resource management. These processes are implemented in planning-based automated systems which are already used in space missions (Muscettola et al. 1998; Chien et al. 2004; Ai-Chang et al. 2004), search and rescue (Hugh et al. 1995), logistics (Tate, Drabble, and Dalton 1996) and many other domains. Since the failure of such systems could have catastrophic consequences, these applications are regarded as safety-critical. Therefore, verification methods that are robust, trustworthy and systematic are crucial to gain confidence in the safety, integrity and correctness of these systems.

The literature is rich with studies on verification of planning systems. For instance, Smith et al. (1999) carried out scenario-based testing and model-based validation of the remote agent that controlled the Deep Space 1 mission. Another example is the verification of the safety of the autonomous science agent design that was deployed on the Earth Orbiter 1 spacecraft (Cichy et al. 2004).

A typical planning system consists of a planning domain model, planning problem, planner, plan, executive, and mon-itor. Planners take as an input a domain model which describes application-specific states and actions, and a problem that specifies the goal and the initial state. From these inputs, a sequence of actions that can achieve the goal starting from the initial state is returned as plan. The plan is then executed by an executive to change the world state to match the desired goal.

Our research focuses on the verification of planning domain models wrt. safety properties. Domain models provide the foundations for planning. They describe real-world actions by capturing their pre-conditions and effects. Due to modelling errors, a domain model might be inconsistent, incomplete, or inaccurate. This could cause the planner to fail in finding a plan or to generate unrealistic plans that will fail to execute in the real world. Moreover, erroneous domain models could lead planners to produce unsafe plans that, when executed, could cause catastrophic consequences in the real world.

This paper addresses the fact that the state-of-the-art verification methods for planning domain models are vulnerable to false positive counterexamples. In particular, unconstrained verification tasks might return counterexamples that are unreachable by planners. Such counterexamples can mislead designers to unnecessarily restrict domain models, thereby potentially blocking valid and possibly necessary behaviours. In addition, false positive counterexamples can lead verification engineers to overlook counterexamples that are reachable by planners.

To overcome these deficiencies, we propose to employ planning goals as constraints during verification. Thus, we introduce *goal-constrained planning domain model verification*, a novel concept that eliminates unreachable counterexamples per se. We formally prove that goal-constrained planning domain model verification of safety properties is guaranteed to return reachable counterexamples if and only if any exist. We also demonstrate two different ways to perform goal-constrained planning domain model verification, one using model checkers and the other using state trajectory constraints planning techniques. To the best of our knowledge, this work is the first to recommend fail-safe planning domain model design practice; introduce the concept of *goal-constrained planning domain model verification*, and demonstrate how model checkers, as well as state trajectory constraints planning techniques, can be used to perform

---

*Supported by [hidden for blind review]

goal-constrained planning domain model verification

The rest of this paper is organised as follows. First, Section 2, contrasts the concepts presented here with related work. Second, Section 3 discusses the problem of unreachable counterexamples in planning domain model verification. Third, Section 4 proposes a design practice for planning domain models that can inherently guarantee domain model safety even in the case of undetected modelling errors. A verification concept of planning domain models that avoids returning unreachable counterexamples is presented in Section 5. Then, Section 6 discusses the implementation of this concept on the Cave Diving planning domain using Spin and MIPS-XXL. Finally, Section 7 concludes the paper and suggests future work.

## 2 Related Work

Closely related, but different, is the work by (Albarghouthi, Baier, and Mcilraith 2009). Their main objective is to treat verification as a planning task, whereas our aim is to demonstrate how model checkers and planners can be used for domain model verification. They proposed to perform system model verification using classical planners. To do this, they first translated the model of the system under verification into a planning domain model. Then, the negation of the safety property to be established, is used as the goal for the planner, which is then consulted to find a plan that acts as counterexample for the given property. In our study, because our aim is to verify domain models against a given property with respect to a specific goal and initial state, we used state trajectory constraints to restrict counterexamples to identify plans that can achieve the planning goal while falsifying the safety property. Unlike (Albarghouthi, Baier, and Mcilraith 2009), where the negation of the safety property is used as the goal, in our verification as planning method, the negation of the safety property is represented as state trajectory constraint and the goal is the given planning goal.

(Raimondi, Pecheur, and Brat 2009) also apply verification as planning to verify planning domain models, starting from LTL specifications. This work fundamentally differs from our work. Raimondi, Pecheur, and Brat (2009) focused their work on translating specification properties into trap formulas which can help in testing the impact of individual atomic propositions on the validity of the overall verified property. However, their method does not consider the interaction between property testing and the original planning goal. Note that finding a planning constraint to exercise a specific atomic proposition is not enough to ensure the constraint itself would be exercised during the planning process. For example, a planning goal might be achieved through a state trajectory that does not exercise the hard constraint used to represent the tested property. Our work is mainly based on investigating this interaction. Therefore, we used state trajectory constraints to guarantee the property is tested while achieving the planning goal. Additionally, their work, just like other similar methods, requires a complete planner to give deterministic results, whereas our work, as discussed in Section 5, guarantees definite verification without this requirement.

(Goldman, Kuter, and Schneider 2012) also used classical planners for planning systems verification, but they examined verifying plans rather than domain models. They proposed an approach that uses classical planners to find counterexamples for a given planning problem and plan instance. Their work and ours are related in that both suggest performing planning verification for a specific planning problem rather than attempting ungrounded verification of a planning system. However, their work is limited to the verification of single plan instances, whereas our method verifies all potential plans that can be spun from a domain model for a specific goal and initial state.

Among others, (Penix, Pecheur, and Havelund 1998; Khatib, Muscettola, and Havelund 2000; Smith et al. 2005; Havelund et al. 2008; Cesta et al. 2010) used model checkers to verify planning domain models. They translated the respective domain models into the input language of the selected model checker. The model checker is then applied to verify the domain model wrt. a given specification property. Similarly, we also proposed a method to verify domain models using model checkers. However, our method differs from the others in two aspects. First, in the way we define the planning domain model verification problem, and, second, in the way we use model checkers to perform verification. As explained in Section 5, we consider the verification of planning domain models to be constrained by a specific goal and initial state pair. In contrast, previous studies perform ungrounded verification of domain models, i.e. leaving the goal and initial state open. As discussed in Section 3, the ungrounded goal and initial state may cause the model checker to return counterexamples that are unreachable when a planner uses the DUV. These unreachable counterexamples can mislead the designers to over-restrict the DUV during the debugging process. On the other side, when the goal and initial state are constrained for verification, then we have shown that the returned counterexamples, if any, are guaranteed to be reachable by any sound planner. The second difference is that, after the planning domain model is translated to the model checker's input language, we augment the model transitions, introducing the negation of the goal as a new constraint that forces the model checker to terminate once the goal is reached. This modification prevents the model checker from returning counterexamples that falsify the given property after satisfying the goal; these are unreachable by planners.

## 3 Unreachable counterexamples in planning domain model verification

Planning domain model verification aims to demonstrate that any produced plan satisfies a set of properties. To achieve this, formal planning domain model verification methods leave the planning goal open. This, we define as *unconstrained* verification of planning domain models, i.e. the verification is expected to hold for any potential goal.

unconstrained verification searches the domain model for a sequence of actions that can falsify the given property, regardless of any other conditions. In particular, whether or not a planner would consider this sequence to be a plan,

is not taken into account. This is a critical oversight, because, when the domain model is used to solve a specific planning problem, the sequence of actions that constitutes such a counterexample might, in fact, be "pruned away" by the planner, if it does not satisfy the planning goal. Therefore, for a specific planning problem, counterexamples that do not achieve the planning goal are deemed unreachable counterexamples from the planner's perspective.

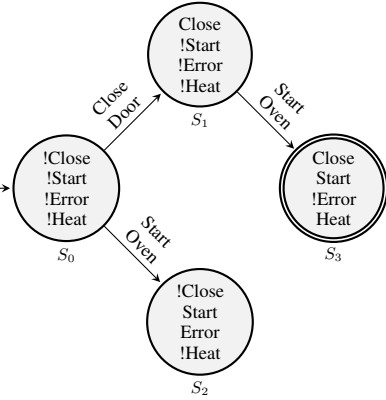

Figure 1: Microwave oven FSM without reachable counterexample

To illustrate this, we use a modified version of the microwave oven example, introduced in (Clarke, Grumberg, and Peled 1999), as presented in Figure 1. A safety requirement would be that the domain model does not allow the generation of erroneous plans, in LTL $p_0 = G(\neg Error)$, where $G$ is the LTL *globally* operator. Unconstrained verification will return $\langle StartOven \rangle$ as a counterexample that when applied to $s_0$ will produce $s_2$ which is an error state. However, when this model is used to find a plan that achieves the goal ($g = Heat$), this sequence will not be considered by the planner as it does not lead to a state that achieves the goal. Moreover, we observe that the valid plan $\langle CloseDoor, StartOven \rangle$ does satisfy the property $p_0$, i.e. is error-free. Thus, the sequence $\langle StartOven \rangle$ from $s_0$ to $s_2$ is an unreachable counterexample for the planner; it does not achieve the goal, nor is it part of a valid plan towards the goal.

Counterexamples that are unreachable by planners exist in the literature. For example, (Smith et al. 2005) used the Spin model checker to verify whether a planning domain model would permit an automated planning system to select plans that would waste resources and therefore not meet the mission's science goals. To express this requirement, they used *"five data-producing activities must be scheduled by any returned plan"* as a property for model checking. The automated system has two data-producing and two data-consuming activities, and a buffer that can hold four data blocks. The goal of the planner is to schedule five data-producing activity instances. The counterexample returned by the model checker represented a plan with the two data-consuming activities scheduled before four data-producing activities. This plan did not contain a fifth data-

producing task, because the data buffer was full after four data-producing activities and the only two data-consuming tasks that would have cleared the buffer, were scheduled at the beginning of the plan with no data in the buffer. Though the model checker found a counterexample to falsify the property, we argue that any sound planner would not generate such a plan, because it does not achieve the planning goal. As such, this counterexample would have been pruned during the planner's goal search, and consequently, it would never have been returned as a plan, i.e. it is unreachable for the planner, yet reachable by a goal-ignorant model checker.

The problem with unreachable counterexamples is that they mislead the designer to unnecessarily restrict the domain model in the process of removing them. Consequently, debugging is made harder and genuine counterexamples could potentially be introduced in the process.

To overcome this, we observe that planning is performed for a specific goal and initial state. To exploit this observation for domain model verification, we propose to use the goal and initial state given to the planner as constraints to ensure that the counterexamples returned by a model checker, or other tools used in this context, falsify the given property while also achieving the planning goal. Thus, instead of performing unconstrained domain model verification, we propose goal-constrained verification of planning domain models. The details of this method are further explained in Section 5. Next, we describe an inherently safe domain model design practice which can help to make domain models safer.

## 4 Inherently safe domain models

The ultimate objective of planning domain model verification is to ensure that the plans produced by the verified domains satisfy a given specification. An alternative and guaranteed way of achieving this goal is to extract plan constraints from the specification, then include them in the domain model. A sound planner using this constrained domain model cannot produce any plan that could violate these constraints. This idea was first noticed in 2005 (Smith et al. 2005) but was dismissed as it was not possible to describe overall plan constraints using PDDL 2.2. However, in 2006 Gerevini and Long (2006) proposed an extension to the PDDL 2.2 language that allows the expression of plan state trajectory constraints. The extended language, called PDDL3.0, was proposed for the fifth international planning competition (IPC-5).

Smith et al. (2005) provided an example of a system consisting of a camera, solid-state recorder and a radio, and a requirement that for all plans, if an image is taken and stored, then it is eventually uplinked. With the hard state trajectory constraints, this property can be expressed as *sometime-after((image is taken and image is stored) image is uplinked)*. With this constraint, any sequence of actions that does not respect this property would not be returned as a plan.

Though including specification properties in the domain model as strong constraints is enough to guarantee that sound planners using the constrained domain models will produce plans that meet the specification, this method will

not be able to find any errors in the domain model. Instead, it will just ensure these errors, if any, are masked and prevented from affecting any plans that could possibly be generated using the modified domain model. As such, this method can be seen as a safety defence layer, a firewall, that prevents any potential property violation. Nevertheless, note that undetected bugs in a domain model could cause what would have been valid plans to be masked, thus unnecessarily restricting the planner. Therefore, further verification efforts are needed to reveal and rectify any underlying errors.

We consider including plan constraints in the domain model to be a good practice to design inherently safe domain models. The effort of extracting formal properties from the specification and inserting them as constraints in the domain model is a small investment in return to the huge benefit of guaranteed safe plans, i.e. plans that are safe "by construction". This, together with our new concept of *goal-constrained* verification, as introduced in the next section, can deliver safe and error-free models.

## 5    Goal-constrained verification of planning domain models

Planning domain model verification covers different objectives, including the domain's correctness, completeness, robustness, effectiveness and safety. The intent of safety verification in this context is to verify that any plan produced from the DUV will satisfy a given safety property. In other words, a domain is considered safe if the domain is guaranteed only to produce plans that satisfy the given safety property when used by a sound planner. This verification task can be performed using advanced search algorithms, such as model checkers or classical planners, to find a valid counterexample for the given safety property.

We define a valid counterexample to be a sequence of actions that, firstly, can falsify the given safety property, secondly, can achieve the planning goal from the given initial state, and, thirdly, none of the sub-sequences of the counterexample can achieve the goal.

Formally, this is defined as follows: Let the planning problem $P$ be a tuple $(D, s_0, g)$, where $D$ is the domain model that describes the set of all available actions $A$, $s_0$ is the initial state and $g$ is the desired goal. $\pi$ is a solution to $P$, a plan, defined as a sequence of actions, where these actions are chosen from $A$. $\pi = \langle a_0, a_1, ..., a_n \rangle$ such that $\pi \models g$ i.e. when $\pi$ is applied to the initial state $s_0$ it yields a sequence of states $S$, $S = \langle s_0, s_1, ..., s_n \rangle$ where the last state $s_n$ satisfies the planning goal $g$, $s_n \models g$. We say a plan $\pi$ satisfies a property $p$, $\pi \models p$, if the sequence of states $S$, generated by the plan $\pi$, satisfies the property $p$, $S \models p$.

Furthermore, as defined in  (Ghallab, Nau, and Traverso 2004), we call a plan $\pi$ a *redundant plan*, if $\pi$ contains a subsequence, $\pi'$, $\pi' \mid \pi$, that achieves the goal $g$.

**Definition 1:** A **valid counterexample** for a safety property, $p$, of a planning problem is a *non-redundant plan*, $\pi$, that falsifies the safety property, $\pi \not\models p$.

Plans are required to be non-redundant in the definition of valid counterexamples to exclude any plans that are enriched with action sequences which are unnecessary to achieve the

planning goal but required to falsify the given safety property. Such plans represent counterexamples that are unreachable by any sound planner when searching for a plan to achieve a given planning problem in a planning task. Such plans represent counterexamples that are unreachable by any sound planner.

To ensure the returned counterexamples are valid, we constrain the verification problem with a goal and initial state, and we exclude any counterexample that is a redundant plan. More formally, the verification problem associated with planning task $P$ is defined as the tuple $V = (D, (s_0, g), p)$. Where $p$ is a formal safety property extracted from a given specification and required to hold over all valid paths that achieve the goal $g$ from the initial state $s_0$.

In this section, we introduced and formally defined the concept of goal-constrained verification of planning domain models. In the following subsections, we demonstrate how this concept can be realized using model checkers and state trajectory constraints planning techniques.

### 5.1    Goal-constrained planning domain model verification using model checkers

Model checkers verify safety properties by searching for counterexamples that falsify those properties. In the case of planning applications, any sequence of actions that does not achieve the given goal, will be pruned by any sound planner. Therefore, in the verification of planning problems, any counterexample that does not achieve the goal of the planning problem should be eliminated on the bases that this counterexample is unreachable by the planner.

To force model checkers to only return valid counterexamples, the safety property is first negated and then joined with the planning goal in a conjunction. This conjunction is then negated and supplied to the model checker as an input property. The final property requires the model checker to find a counterexample that both, falsifies the safety property and satisfies the planning goal. Note that, unlike Def. 1, this permits sequences that falsify the property after satisfying the goal. However, once the goal is achieved, planners terminate the search, thus rendering such sequences unreachable. To eliminate these sequences, model transitions should be augmented with an additional guard, representing the negation of the goal, to restrict all transitions once the goal is achieved. With this modification, the model checker is forced to return counterexamples that falsify the safety property before achieving the goal, because once the goal is satisfied no further transitions will be permitted.

For a verification problem $V = (D, (s_0, g), p)$ we first translate the domain model $D$ into the model checker's input language, obtaining the model $M$ that incorporates the initial state $s_0$. Then, a model checker is applied to the verification problem $V' = (M, \neg F(g))$ to establish that

$$\exists \pi. \ \pi \models F(g), \tag{1}$$

where $F$ is the LTL *eventually* operator.

The model $M$ is modified to $M'$ by augmenting the guards of all transitions with the negation of the goal condition. The model checker is then applied to the verification

problem $V'' = (M', p')$ where $p'$ is defined as follows:

$$p' = \neg\big(F(\neg p) \wedge F(g)\big) \tag{2}$$

There are two possible outcomes of the verification task $V''$. If the model checker returns a counterexample, $\pi$, then:

$$\pi \not\models p' \tag{3}$$
$$\equiv \pi \models (F(\neg p) \wedge F(g)) \tag{4}$$

From the definition of the LTL *eventually* operator $F$:

$$\exists i \geq 0, \; s_i \in S, s_i \models \neg p \tag{5}$$
$$\exists j \geq 0, \; s_j \in S, s_j \models g \tag{6}$$

It follows that there is at least one sequence $S$ that falsifies the property $p$, and there is a state $s_j$ in that sequence which satisfies the goal $g$, according to (5) and (6). In addition to that, in the sequence $S$, $p$ is guaranteed to be falsified before $g$ is satisfied, due to the modification we introduced in the model $M'$. Thus, the plan $\pi$ is a valid counterexample for the original safety property $p$ as per Def. 1. This proves that the DUV does not satisfy the safety property $p$ with respect to the goal and initial state.

The other potential outcome is that the model checker fails to find a counterexample, then for all plans, $\pi$:

$$\pi \models p' \tag{7}$$
$$\equiv \pi \not\models (F(\neg p) \wedge F(g)) \tag{8}$$

It follows that $\pi \not\models F(\neg p) \vee \pi \not\models F(g)$. Furthermore, from (1) we know that $\exists \pi. \; \pi \models F(g)$. Therefore for all plans, $\pi$:

$$\pi \not\models F(\neg p) \tag{9}$$
$$\equiv \pi \models \neg(F(\neg p)) \tag{10}$$
$$\equiv \pi \models G(p) \tag{11}$$

That means $p$ is always true for all possible plans. Which proves that the DUV satisfies the original property with respect to the goal and initial state.

## 5.2 Goal-constrained planning domain model verification using planning techniques

Domain models can be verified to only produce valid plans, in terms of satisfying given a property, for a specific goal and initial state pair using sound planners. This is achieved by consulting the planner over the DUV to produce a plan that can satisfy the goal and the negation of the property. If the domain model permits producing plans that, along with achieving the goal, contradict the safety property, then an unsafe plan can be found. Thus, the returned plan is a counterexample that demonstrates that the safety property does not hold. On the other hand, if the domain model does not permit the generation of plans that can satisfy the negation of the safety property while achieving the goal, then the planner will fail. Thus, the property holds in any plan produced for the given goal.

A benefit of goal-constrained planning domain verification is, where a planner is used to perform the verification

task, there is no need to for this planner to be complete, as long as the planner used for the verification is also the planner that will be used during the planning task. This is because any counterexample not found by that planner during verification, will then also not be reached by the same planner during the planning task.

The following subsection provides a description of how state trajectory constraints can be used to verify planning domain models for a specific goal and initial state.

**Goal-constrained planning domain verification using planning techniques with state trajectory constraints** The PDDL3.0 state trajectory constraints, first mentioned in Section 4, can be used to perform planning domain model verification. First, the negation of the given property is expressed using PDDL3.0 modal operators and embedded in the original domain model as state trajectory constraint. The modified model is then used by a planner, as described earlier, to perform the verification.

For a verification problem $V = (D, (s_0, g), p)$, we first apply a planner to the planning problem $P = (D, s_0, g)$ to establish that there is a plan that solves $P$

$$\exists \pi. \; \pi \models g. \tag{12}$$

Then, the safety property $p$ is negated and expressed in terms of PDDL3.0 modal operators as shown in (Gerevini et al. 2009). The result is added as state trajectory constraint to the original domain model.

Using the algorithm proposed in (Edelkamp, Jabbar, and Nazih 2006), the new model is transformed into a PDDL2 compatible version. This is performed by first translating the state trajectory constraint into a non-deterministic finite state automaton (NFA) which can monitor property violations by inserting additional predicates and actions conditional effects into the model to simulate and observe the behaviour of the automaton that represents the constraint.

This yields a new planning problem $P' = (D', s_0', g')$, where $D'$, $s_0'$, $g'$ are modified instances of $D$, $s_0$, $g$ that are supplemented with the additional predicates and actions conditional effects of the automaton that represents the introduced constraint. Then, a planner is applied to $P'$ with two possible outcomes. If the planner finds a plan then:

$$\exists \pi. \; \pi \models g' \tag{13}$$

Since the satisfaction of $g'$ implies both, the satisfaction of the original goal $g$ at the last state of the sequence $S$, and the satisfaction of the state trajectory constraint by the sequence $S$, (13) can be rewritten as

$$\exists \pi. \; \pi \models g, \pi \models \neg p. \tag{14}$$

Furthermore, from (14) it follows that $\pi \not\models p$, confirming that there is at least one plan that achieves the goal while not respecting the safety property. Therefore, this plan is a valid counterexample for that property as per Def. 1. Hence, the DUV does not satisfy the property wrt. the planning goal and initial state.

Alternatively, if the planner fails to find a plan, then, as

opposed to (14), we have

$$\nexists \pi. \ (\pi \models g \land \pi \models \neg p) \tag{15}$$

$$\equiv \forall \pi. \ \neg(\pi \models g \land \pi \models \neg p) \tag{16}$$

$$\equiv \forall \pi. \ (\pi \not\models g \lor \pi \not\models \neg p) \tag{17}$$

Given (12), (17) can be simplified to:

$$\forall \pi. \ \pi \not\models \neg p \equiv \ \forall \pi. \ \pi \models p \tag{18}$$

Hence, all plans satisfy the safety property. Therefore, the property holds for the planning domain model wrt. the given goal and initial state.

## 6 Example

In this section, we discuss how goal-constrained planning domain verification can verify safety properties using both the Spin model checker (Holzmann 2004) and the MIPS-XXL planner (Edelkamp, Jabbar, and Nazih 2006). We perform constrained and unconstrained verification tasks to show how unlike the latter task our method does not return unreachable counterexamples. As an example, we consider the classical cave diving planning domain taken from the Satisfying Track of the IPC-2014 (IPC2014 2014). The problem consists of an underwater cave system with a finite number of partially interconnected locations. Divers can enter the cave from a specific location, entrance, and swim from one location to an adjacent connected one. They can hold up to four oxygen tanks and they consume one for every swim and take-photo action. Only one diver can be in the cave at a time. Finally, divers have to perform a decompression manoeuvre to go to the surface and this can be done only at the entrance. Additionally, divers can drop tanks in or take tanks from any location if they hold at least one tank or there is one tank available at the location respectively.

The planning goals of this domain, as provided in the problem files in the IPC-2014, consist of two parts. The first part dictates the required underwater location of which the photo is to be taken (we call it mission target) and the second part which mandates the divers should return to the surface after completing the mission (we call it safety target).

A critical safety property is that divers should not drown i.e. they should not be in an underwater location, other than the entrance, where neither the divers nor the location has one full oxygen tank at least.

To enable the planner and the model checkers to explore the entire state space, we simplified this domain by ignoring the "precludes" condition from the original domain as it does not affect the verification of the drowning safety property. Consequently, we considered only one diver and we modified some actions to enable the diver to go back into the water after a dive. These modifications are further explained in the commented simplified planning domain model PDDL file which is provided along with the tasks problem PDDL and Promela files online [address hidden for blind review].

First, we translated the planning domain model from PDDL to Promela. Thus, the verification results using the translated model only hold provided that the translation is valid. The verification of the translation is outside the scope and focus of this paper and left for future work.

In this example, the chosen planning goal is to have a photo of the first location, $L_1$, and to get the diver outside the water. The verification tasks are:

1 - Unconstrained verification with only the safety property: Both Spin and MIPS-XXL found a counterexample $\langle$prepare-tank, enter-water, swim($L_0, L_1$)$\rangle$. Indeed, this counterexample leads the diver to a drowning state. At the end of this sequence, the diver will be in underwater location $L_1$ which is not the entrance so they can not surface and with no oxygen tank to swim back to the entrance. However, this is not a plan because it does not achieve any useful goal. Therefore, it will not be produced by any sound planner when it is used in a practical scenario (taking a photo of any location).

2- Verification with safety property and incomplete goal (mission target only): Both Spin and MIPS-XXL returned $\langle$prepare-tank, prepare-tank, enter-water, swim($L_0, L_1$), take-photo$\rangle$. This counterexample achieves the goal and violates the property. However, without the safety part of the goal, it would be possible to generate plans that imply divers should swim to an underwater location and take a photo of it without requiring the divers to return to the surface. These kind of plans are illegal as they do not respect the safety part of the goal. Therefore, such sequences are unreachable counterexamples i.e. will never be produced by any sound planner while planning for a legal goal.

3- Verification using Spin with both safety property and proper goal but without the augmented model $M'$: Spin found a counterexample $\langle$prepare-tank, prepare-tank, prepare-tank, prepare-tank, enter-water, swim($L_0, L_1$), take-photo, swim($L_1, L_0$), decompress, enter water, swim($L_0, L_1$)$\rangle$. This counterexample achieves the goal and violates the safety property but only after the goal is achieved. Therefore, this is also an unreachable counterexample because a sound planner will terminate after achieving the goal and any counterexample that violates the property after achieving the goal will not be returned. Hence, it is unreachable.

4- Goal-constrained planning domain verification, as presented in this paper, the result was: No plan is returned by the planner MIPS-XXL with complete exploration and no counterexample is returned by Spin with exhaustive verification mode. This means the planning domain model has no provision of producing a plan that can violate the safety property before achieving the goal. I.e. this model is safe with respect to the given property and goal pair.

Though the counterexamples returned by the incomplete verification tasks number one, two and three are obviously unreachable and should not misguide the designers to overcomplicate the model, in a real world sized application such unreachable counterexamples can be critical and much more diffcult to recognise and avoid. We expect that our proposed concept can save practitioners a huge amount of person-hours trying to alter planning domain models for behaviours that their planners will never experience in practice.

## 7 Conclusions and future work

The verification of planning domain models is essential to guarantee the safety of planning-based automated systems.

Unreachable counterexamples returned by unconstrained planning domain model verification techniques undermine the verification results.

In this paper, we have discussed the potential deficiencies of this problem and provided an example of an unreachable counterexample form the literature. We then introduced goal-constrained verification, a new concept to address this problem, which restricts the verification task to a specific goal and initial state pair. This limits counterexamples to those practically reachable by a planner that is tasked with achieving the goal given the initial state. Consequently, our method verifies the domain model only wrt. a specific goal and initial state. This is an acceptable limitation, given that planners also operate on this basis.

We have demonstrated how model checkers and planning techniques can be used to perform goal-constrained planning domain model verification. In addition, we have recommended an inherently safe practice for domain model design that guarantees the safety of domain models "by construction" in case of undetected modelling errors. Goal-constrained domain model verification ensures accurate verification results and complements the inherently safe domain model design practice to generate safe and error-free planning domain models.

In conclusion, the main message of this paper is that the direct application of verification algorithms to the planning domain model verification problem can return counterexamples that would never be reached by planners in real planning tasks. These unreachable counterexamples can mislead the designers to perform unnecessary remediations that can be prone to errors. The proposed solution is simple which makes it readily usable in practice. It is also effective as formally proven in the paper.

Currently, we are investigating the use of Temporally Extended Goals (TEGs) translators (Torres and Baier 2015) to perform goal-constrained domain model verification. As future work, we intend to automate the proposed methods, so that they can be applied to real-world sized planning domain models. Finally, we would like to perform an empirical comparison of the proposed methods to assess their scalability and performance.

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
