# OpenReview forum: "Goal-constrained planning domain model formal verification of safety properties"
_icaps-conference.org/ICAPS/2019/Workshop/KEPS — KEPS 2019_

### Official Review · AnonReviewer2 · 2019-05-08
**Addressing false positives in domain model verification**

**Rating:** 3
**Confidence:** 2

**Review:**

Planning domain model verification is an important part of Knowledge Engineering (KE) process. The paper presents a domain model verification technique that avoids unreachable counter-example plans (i.e., false positives) by considering an initial state and a goal.

The work goes in an interesting direction as false positives, as the authors argue, might mislead engineers who develop planning domain models. The limitation of the work is that it requires problem instances which might be difficult to obtain such that they are representative of the class of the problems one wants to solve.

The paper presents three ways how the verification can be done, model checkers, classical planning and planning with trajectory constraints. The evaluation is done on the Cave Diving domain and a comparison to state-of-the-art is made. The evaluation is however somewhat preliminary as it does not even report how long (CPU-time) the verification process takes. Also, some examples that have actual errors (true positives) could have been included.

Minor comment:

In many occasions, the citations, when the reference authors stand for a subject in the sentence, are in an incorrect format. For example,  "(Smith et al. 2005) used the Spin model checker" should be written as " Smith et al. (2005) used the Spin model checker"